# Multi-Stage Transcriptome Analysis Revealed the Growth Mechanism of Feathers and Hair Follicles during Induction Molting by Fasting in the Late Stage of Egg Laying

**DOI:** 10.3390/biology12101345

**Published:** 2023-10-19

**Authors:** Lujie Zhang, Chunxia Cai, Xinxin Liu, Xiaoran Zhang, Zhiyuan An, Enyou Zhou, Jianzeng Li, Zhuanjian Li, Wenting Li, Guirong Sun, Guoxi Li, Xiangtao Kang, Ruili Han, Ruirui Jiang

**Affiliations:** 1The Shennong Laboratory, Zhengzhou 450002, China; zlj18236117037@163.com (L.Z.); ccx17837161772@163.com (C.C.); lxinxin1220@163.com (X.L.); liwenting_5959@hotmail.com (W.L.); grsun2000@126.com (G.S.); liguoxi0914@126.com (G.L.); xtkang2001@263.net (X.K.); 2College of Animal Science and Technology, Henan Agricultural University, Zhengzhou 450046, China; zhangxiaoran6677@163.com (X.Z.); anzhiyuan0076@163.com (Z.A.); 18638621293@163.com (E.Z.); lijianzeng1205@163.com (J.L.); lizhuanjian@163.com (Z.L.)

**Keywords:** induced molting, feathers, hair follicles, regeneration, transcriptome, thyroid hormones

## Abstract

**Simple Summary:**

Feather replacement is one of the most typical features of fasting-induced physiological remodeling, but the specific mechanism is unknown. By observing the changes in feathers and hair follicles throughout the process, this study reveals that molting can increase the market value of culled laying hens and improve the carcass appearance of chilled chickens. In addition, combined with transcriptome sequencing, candidate genes related to hair follicle development were found, namely *DSP*, *CDH1*, *PKP1*, etc., and a specific pathway elucidating how thyroid hormone affects feathering was proposed. These data provide a valuable resource for the analysis of the molecular mechanisms underlying the cyclical growth of hair follicles in the skin during induced molting.

**Abstract:**

Induced molting is a common method to obtain a new life in laying hens, in which periodic changes in feathers are the prominent feature. Nevertheless, its precise molecular mechanism remains unclear. In this study, feather and hair follicle samples were collected during fasting-induced physiological remodeling for hematoxylin–eosin staining, hormone changes and follicle traits, and transcriptome sequencing. Feather shedding was observed in F13 to R25, while newborns were observed in R3 to R32. Triiodothyronine and tetraiodothyronine were significantly elevated during feather shedding. The calcium content was significantly higher, and the ash content was significantly lower after the changeover. The determination of hair follicle traits revealed an increasing trend in pore density and a decrease in pore diameter after the resumption of feeding. According to RNA-seq results, several core genes were identified, including *DSP*, *CDH1*, *PKP1*, and *PPCKB*, which may have an impact on hair follicle growth. The focus was to discover that starvation may trigger changes in thyroid hormones, which in turn regulate feather molting through thyroid hormone synthesis, calcium signaling, and thyroid hormone signaling pathways. These data provide a valuable resource for the analysis of the molecular mechanisms underlying the cyclical growth of hair follicles in the skin during induced molting.

## 1. Introduction

Molting is an inherent biological process in avian species, while induced molting has advantages such as enhanced egg production, cost efficiency in rearing, and adaptability to market demands, when compared to natural molting [1]. Fasting-induced molting (IM) has emerged as the prevalent method for artificially stimulating feather replacement worldwide. This method has gained popularity due to its simplicity and improved economic benefits [2]. One of the prominent features of this process is feather variation. However, molting is not a mere feather alteration process; it is a complex physiological process that is tightly regulated by both internal and external stimuli. Research has shown that a multitude of factors, including seasonal variations, food intake, hormones, and others, all collectively contribute to the molting process [3]. Feather development starts in the hair follicle (HF), which is a complex mini-organ composed of epidermal and mesenchymal (dermal) components. The HF regulates feather growth, replacement, and morphological structure [4]. The entire process of hair follicle growth, including tract formation, bud growth, and recirculation processes, has been comprehensively described [5]. Nevertheless, there is limited research available regarding the patterns of changes observed in both feathers and HF during IM.

Starvation stimulation controls hormone secretion from the anterior pituitary gland, impacting the release of thyroid-stimulating hormone (TSH), adrenocorticotropic hormone (ACTH), and gonadotropin-releasing hormone (GnRH) [6]. Studies have shown that prolactin can inhibit hair follicle growth [7]. Similarly, luteinizing hormones and steroid hormones (testosterone, estradiol, and progesterone) were found to be associated with feather changes. Specifically, progesterone (P4) levels decreased during the growth of new feathers [8]. Meanwhile, the adrenocorticotropic hormone was also increased during natural molting [9]. Additionally, triiodothyronine (T3) and tetraiodothyronine (T4) play a role in somatic cell evolution and influence the formation of feathers. Increased plasma levels of T4 have been found to trigger molting and promote the growth of new feathers [10,11]. The hormone content is closely related to feather variation, and hair follicle development undergoes a cyclic cycle of prophase, regression, and quiescence. This process is regulated by several genes, including factors that both promote and inhibit the action of hair follicle growth factors [12].

With the development of next-generation sequencing technology, RNA sequencing (RNA-seq) technology has been widely used to reveal the genetic mechanisms underlying normal chicken skin color, as well as the development of embryonic and adult feathers [13,14,15]. However, few researchers have focused on feather and hair follicle changes during IM. The transcriptome was routinely enriched through Gene Ontology (GO) and Kyoto Encyclopedia of Genes and Genomes (KEGG) analyses, while the series test of cluster analysis and weighted correlation network analysis (WGCNA) can include other less commonly studied regulatory molecules that may play a role in different signaling pathways.

Therefore, this study focuses on the growth pattern of feathers and hair follicles during IM and aims to evaluate the effect of forced feather change by second egg-laying feather recovery. RNA-seq binding phenotypes were conducted to identify the signaling pathways and different genes that may contribute to altered hair follicle traits.

## 2. Materials and Methods

### 2.1. Experimental Animals

The experiment was carried out at the poultry germplasm resource field of Henan Agricultural University. A total of 100 Houdan females, aged 320 days, were randomly selected. The protocol followed in this study was approved by the Institutional Animal Care and Use Committee of China (IACUC) to minimize animal suffering.

### 2.2. Animal Experimental Design

Sick, frail, lean, obese, and mentally depressed chickens were removed from the flock before the trial began. This was carried out to ensure that the replacement flock was homogeneous and in good overall condition. Thirty chickens were randomly selected for regular weighing, with measurements taken every three days. The weight loss rate was recorded and closely monitored throughout the trial. Feeding resumed when the marked individuals reached a weight loss of 30%, which occurred on the 15th day. The specific implementation plan for inducing feather change in the flock is described in Table 1.

### 2.3. Sample Collection

The changes in the feathers during molt induction were recorded. After anesthesia with 0.2% concentration of pentobarbital (40 mg/kg) at specific time points, a total of 8 chickens were euthanized (cervical dislocation): F0 (on the day before the first day of feed breaking), F7 (on the 7th day of feed breaking), F15 (on the 15th day of feed breaking), R5 (on the 5th day of feed resuming), R10 (on the 10th day of feed resuming), R15 (on the 15th day of feed resuming), R20 (on the 20th day of feed resuming), R25 (on the 25th day of feed resuming), and R32 (on the 32nd day of feed resuming). Blood was collected, and serum was separated to determine the levels of P4, prolactin (PRL), glucocorticoid (GC), adrenaline (AD), T3, and T4 using enzyme-linked immunosorbent assay (ELISA). The first nine periods involved plucking the primary flight feathers in the axial feather direction. The lengths of these feathers were then measured, and the average value was calculated. Skin tissues from the same parts of the neck, chest, back, legs, and wings were collected. A portion of the tissue was placed in RNA-free Eppendorf for RNA extraction, while another portion was immersed in 4% formaldehyde for hair follicle characterization and sectioning.

### 2.4. Paraffin Tissue Embedding and Section Manufacture and HE Staining

Tissue samples (0.5 × 0.5 mm) were fixed in a 4% formaldehyde solution for 24 h and underwent a series of processes, including dehydration, clearing, and paraffin embedding. The embedded sample was then sliced, dried, dewaxed, dehydrated, and stained with hematoxylin–eosin. Finally, the prepared slides were observed under a microscope.

### 2.5. Feather Composition Determination

The moisture, crude fat, crude protein, crude ash, calcium, and phosphorus contents of feathers were determined concerning GB/T 6435-2014, GB/T 6433-2006, GB/T 6432-2018 methods, using GB/T 6438-2007, GB/T 6436-2018 and GB/T 6437-2018 standards, respectively.

### 2.6. Determination of Skin Follicle Properties

The thickness (cm) and pore diameter (mm) of the 4% formaldehyde-fixed skin samples were measured using vernier calipers. Each sample was randomly measured at three locations, and the average value was recorded. For each sample, three hair follicles were randomly selected, and their measurements were taken three times. The average value of these measurements was then calculated. Pore density was determined by randomly selecting three locations within the 1.5 × 1.5 cm^2^ model, counting the number of pores at each location, and taking the average value.

### 2.7. RNA Extraction and RNA-Seq Analysis

#### 2.7.1. Quality Control of Sequencing Data

After extracting F0, F15, R5, and R32 four periods (three samples per period) of back skin tissue total RNA using Trizol reagent from Invitrogen, the RNA quality was assessed using an Agilent 2100 Bioanalyzer from Agilent Technologies as well as RNase-free agarose gel electrophoresis. After total RNA extraction, reverse transcription and double-stranded DNA synthesis were performed to obtain the transcriptome library. To ensure library quality, PCR amplification, library concentration detection, and library fragment size analysis were conducted. Then, sequencing was performed using the Illumina HiSeq NOVAseq 6000 platform provided by Genedenovo Biotechnology (Guangzhou, China). The RNA-seq sequencing data were deposited into NCBI (PRJNA988236).

#### 2.7.2. Raw Data Quality Control, Comparison with Reference Genome, and DEG Analysis

The raw data contained low-quality and invalid data, which would interfere with subsequent analysis. Therefore, it was necessary to use FASTQ for quality control to filter out the low-quality data and obtain clean reads. We removed low-quality reads including all A bases and reads that had adapters, N ratios greater than 10%, or more than 50% low-mass bases (Q ≤ 20) [16]. The reads after quality inspection were compared with the chickens’ GRCg7b version rRNA (http://asia.ensembl.org/Gallus_gallus/Info/Index) (accessed on 31 December 2022) using HISAT2. Using DESeq2 software [17], the obtained *p*-value was corrected using the Benjamini–Hochberg method, and the differentially expressed gene met the two standard conditions of having a false discovery rate (FDR) parameter less than 0.05 and absolute fold change ≥2.

#### 2.7.3. GO and KEGG

Hypergeometric tests were performed to identify significantly enriched Gene Ontology (GO) terms in differential genes compared with the whole genomic background. The enrichment analysis of KEGG pathways was conducted using the KEGG pathway as a unit, and hypergeometric tests were carried out to identify pathways that showed significant enrichment in different genes compared with the background gene set.

#### 2.7.4. Series Test of Cluster

Using the STEM software, gene expression data were pre-processed using log2 normalization to generate the default selection of the 20 most representative modules. For this analysis, *p*-values were utilized to assess the number of genes within each module relative to the expected value from a random distribution. A smaller p-value indicates a greater significance of the gene set, suggesting that more genes exhibit a specific pattern of variation due to sample variation.

#### 2.7.5. WGCNA

The phenotypic data were integrated with transcriptomic data for WGCNA using the platform provided by Genedenovo Biotechnology Co. (Guangzhou, China).

#### 2.7.6. PPI

The DEGs associated with skin developmental functions were screened through a combination of GO and KEGG analyses. Protein interactions were predicted using the STRING database, and all interactions were filtered with a minimum confidence score of 0.7. The PPI network maps were then generated using Cytoscape (version 3.9.1).

#### 2.7.7. Validation of RNA-Seq Data Using Quantitative Real-Time PCR (qRT-PCR)

To ensure the accuracy and reliability of the RNA-seq results, the same RNA samples that were used for sequencing were utilized as templates. From the sequencing results, DEGS were randomly selected for validation via qRT-PCR. The primers were designed based on the gene sequences published on NCBI, and they were synthesized by Shangya Science Biotechnology. The primer sequences are provided in Table 2. The target gene was quantified using chicken GAPDH as an endogenous control using a standard gene, in which six samples were selected from each tissue, and two technical replicates were performed for each sample. The PCR reaction system consisted of 10 μL, including 5 μL of 2× SYBR Green Mix, 0.5 μL each of upstream and downstream primers, 1 μL of cDNA template, and 3 μL of double-distilled water. The PCR reaction procedure consisted of a pre-denaturation step at 95 °C for 5 min, followed by denaturation at 95 °C for 30 s, annealing at 60 °C for 30 s, and extension at 72 °C for 30 s, with a total of 35 cycles.

### 2.8. Data Processing

The raw data obtained from the quantitative PCR analysis were analyzed using the relative quantification 2^−ΔΔCt^ calculation method to determine the gene expression levels during different periods in the chicken HF. The data were then visualized using GraphPad Prism 9.0. For statistical analysis, an independent sample *t*-test was performed in SPSS 26.0, with *p* < 0.05 indicating significant differences and *p* < 0.01 indicating highly significant differences. The results are presented as the mean ± standard error.

## 3. Results

### 3.1. Changes in the Flock during IM

#### 3.1.1. Chicken Status and Feather Characteristics at Different Periods of IM

During the onset of fasting, the chickens displayed restlessness and engaged in behaviors such as egg pecking. Additionally, some chickens ceased laying eggs, and by the seventh day of fasting (F7), all laying activity had ceased. Subsequently, the chickens entered a tranquil and resting state. After F15, the feeding of the chickens resumed. The chickens exhibited an active mental state, and by the R20, they began to recommence egg production, displaying a similar state as before the fasting period.

As depicted in Figure 1, chicken feathers underwent varying degrees of damage and shedding before fasting (Figure 1A). After the fasting period and starting the resumption of feeding, a phenomenon of feather replacement occurred, including the replacement of main wing feathers (Figure 2A). The shedding of chicken feathers commenced during the F13 period, reached 50% during the R6 period, reached 100% during R2, and ceased by the R25 period. During the R10 period (Figure 1B), new feather buds started to grow. In the beginning and growth stages of feather development (Figure 1C), the feather branches became thick and full, the feather medulla became visible, and the skin tissue became more closely integrated. During R32 (Figure 1D), the feathers were fully grown and covered the chicken completely. In the R47 period (Figure 1E), the feathers reached maturity and entered the resting stage. During this stage, the feather branches became shrunken and refined. The feather shafts were hollow and transparent, devoid of pith fluid (Figure 1F). Additionally, the blood vessels at the base of the feathers shrank and were loosely combined with the skin tissue.

As observed in Figure 2A, the shedding of chicken feathers initiated in the late-laying hens starting from F13. The feather loss rate reached 50% during R6, and it reached 100% during R12. Feather shedding completely ceased by the R25 period.

#### 3.1.2. Regeneration of Feathers in Different Parts of the Houdan Chicken

As depicted in Figure 2B, the neck feathers were the first to develop feather buds at R3. By R5, they reached 50% growth and achieved full growth (100%) at R9. Subsequently, the chest feathers commenced developing feather buds at R5 and completed growth (100%) at R15. Feather growth of the back, legs, and wings started simultaneously at R7. However, the legs exhibited faster feather growth compared with the back and wings. The wings followed suit, and the back exhibited the slowest growth rate. All three regions reached full growth (100%) at R17. Prior to molting, as shown in Figure 2D (F0), exposed dorsal skin was observed, with the emergence of feather buds during R11. Subsequently, during R15, body feathers continued to grow, and eventually, a complete feather coverage was reached at R22.

#### 3.1.3. The Replacement of the Overall Primary Flight Feathers of the Chicken Flock

The results are presented in Figure 2C. Throughout the fasting period, there was a tendency toward an increase in the length of the main wing feathers (Figure 2Ca). After the resumption of feeding, there were significant changes in the length of the main wing feathers between R10 and R25, together with an increased number of main wing feathers that underwent shedding (Figure 2Cb). By R32, the shedding of main wing feathers ceased, and their length returned to the pre-fasting state (Figure 2Cc).

### 3.2. Serum Hormone Changes and Histological Observation

#### Changes in Hormone Levels in the Serum

The results of changes in the hormone content in serum at different time points showed that the levels of P4 and PRL decreased as fasting was prolonged. Specifically, the level of P4 reached significance at F15 (*p* < 0.05), while PRL was significantly downregulated at F7 (*p* < 0.05). During the resumption of feed intake, the levels of P4 and PRL hormones continuously declined, while they were specifically decreased at R20. As feed intake gradually resumed, there was a significant increase in both hormones at R47, reaching the levels observed during the early fasting period (Figure 3A). During the fasting period, there was no significant change in the GC content. However, upon the resumption of food intake, the GC content gradually increased. Specifically, R20 and R25 showed significantly higher GC content than R5 (*p* < 0.01). After the resumption of feeding, the GC content tended to return to pre-fasting levels. Additionally, AD levels increased during both the fasting period and resumption of feeding intake, with R32 and R47 gradually approaching pre-fasting levels. It is worth noting that, during R10, a peculiar elevation in the GC content was observed, while AD decreased (Figure 3B). During the fasting period, and at the start of the resumption of feed intake, the levels of T3 and T4 increased, reaching significant levels at R20 (*p* < 0.01). This was accompanied by a decrease in T3 and T4 levels from R32 to R47, returning to pre-fasting levels, as shown in Figure 3C.

### 3.3. Histological Observations

The results of HE staining are shown in Figure 3D. The diameter of the HF gradually increased from R5. During the F0 period, HF was in the resting stage, and the feathers shed. In the F15 period, HF entered the regressive stage. In the R5, new HF started to grow, indicating the anagen stage with maximum diameter. Subsequently, the diameter of the HF gradually decreased from R10 to R25, indicating entry into the regressive stage.

### 3.4. Feather Composition Measurement

The changes in the nutrient content of feathers during the four periods are depicted in Figure 3E. The moisture content decreased during the fasting period, with F15 showing a significant decrease compared with all other periods (*p* < 0.05). After the resumption of feeding, the moisture content increased and reached the level of F0. The crude protein levels increased during F15 compared with F0 and R32 (*p* < 0.01). Subsequently, the levels decreased and returned to the F0 level. In contrast, the crude ash and calcium content of the feathers exhibited a decreasing trend, with R32 significantly lower than F0 (*p* < 0.05). The levels of phosphorus and crude fat did not show significant differences across the different periods.

### 3.5. Skin Follicle Properties

The results presented in Figure 3F indicate that the dorsal skin thickness decreased during the fasting period. Specifically, F15 exhibited a significant decrease (*p* < 0.05, *p <* 0.01), with a tendency to return to pre-fasting levels after the resumption of feeding. Moreover, there was a trend of increase in dorsal pore density, with R5 and R32 showing significantly higher values than F0 and F15 (*p* < 0.05). Conversely, the dorsal pore diameter exhibited a decrease, with the diameter significantly higher during F0 and F15 than during R5 and R32 (*p* < 0.05).

### 3.6. Transcriptome Analysis of Late-Laying Chickens’ Skin

#### 3.6.1. Quality Control of Sequencing Data

After data were filtered, a total of 37586978bp of clean data were obtained. The Q20 of each sample’s clean read was greater than 97.00%, Q30 was greater than 92.55%, and the GC content was in the range of 48.35~50.53% (Appendix A). The quality control results showed that the sequencing results were reliable and adequate for further data processing.

#### 3.6.2. Differential Expression Gene (DEG) Analysis

A total of 1374 upregulated genes and 1079 downregulated genes were identified (Figure 4). The majority of downregulated genes were observed in F0-VS-F15, F0-VS-R5, and R5-VS-R32, while the majority of upregulated genes were found in the F0-VS-R32, F15-VS-R5, and F15-VS-R32 groups. The log2(FC), *p*-value, and FDR values of the top ten genes in different comparison groups are shown in Appendix A.

#### 3.6.3. GO and KEGG Analyses

GO and KEGG functional enrichment analyses were performed on the DEGs. GO enrichment analysis pointed to the establishment of the skin barrier, the regulation of water loss via the skin, biological adhesion, cell adhesion, and epidermal cell differentiation in the F0-VS-R32 group (Figure 5A). GO factors such as skin development, the establishment of the skin barrier, the regulation of water loss via skin were observed in the F15-VS-R32 (Figure 5B) group. KEGG enrichment analysis revealed that cell adhesion molecules and VEGF signaling pathway were significantly enriched in the F0-VS-R32 group (Figure 5C), and Vitamin B6 metabolism was significantly enriched in the F0-VS-R5 group (Figure 5D).

#### 3.6.4. Series Test of Cluster

The series test of cluster analysis was performed for all target differential genes in the 20 models obtained (Figure 6A). Statistically significant color modules, namely modules 19, 17, 16, 13, 14, 4, 12, and 11 (*p* < 0.05), were identified. Module 19, consisting of 366 DEGs, exhibited an increasing trend, as shown in Figure 6B. On the other hand, module 17, comprising 622 DEGs, showed an initial increase followed by a stable level (Figure 6C). To further analyze the data, the differentially expressed genes of the two modules were subjected to GO functional annotation. Module 19 was found to be mainly involved in biological processes such as keratinization, multicellular water homeostasis, lipid metabolism, water homeostasis, the establishment of the skin barrier, cellular lipid metabolic processes, epidermal cell differentiation, and the regulation of cellular water loss (Figure 6D). On the other hand, module 17 primarily exhibited biological processes related to inner ear development, epidermal development, epithelial development, the regulation of body fluid levels, epidermal cell differentiation, epithelial cell proliferation, the establishment of the skin barrier, cell adhesion, and adhesion (Figure 6E).

#### 3.6.5. Protein–Protein Interaction (PPI) Network Analysis

The target gene sets derived from the differential analysis and series test of cluster analysis were separately imported into the STRING database for PPI network analysis. The visualization results of the PPI network analysis were generated using Cytoscape 3.9.1 software. The top ten pivotal genes for difference analysis were *DSP*, *FA2H*, *CERS3*, *FST*, *DHCR24*, *Krt75*, *SPTLC3*, *ELOVL7*, *ELOVL47*, and *Krt6a* (Figure 7A). The top ten predicate genes for the series test of cluster analysis were *CDH1*, *PKP1*, *Pkp3*, *DSP*, *EPHA1*, *DSC1*, *CadN*, *TCF7*, *DSG27*, and *TIAM1* (Figure 7B). Among these genes, DSP was the gene that they jointly screened.

#### 3.6.6. WGCNA

WGCNA divided tens of thousands of genes into 20 modules (color-coded) with similar expression patterns, as shown in a dendrogram (Figure 8A). In the dendrogram, each branch represents a module, and each leaf within the branch represents a gene. Figure 8B illustrates the number of genes present in different modules. Correlation analysis was conducted using module eigenvalues and specific trait data to identify the modules most relevant to the phenotype. Thirteen modules exhibited significant correlations with the phenotype, with the MM.TAN, MM.YELLOWGREEN, and MM.DARKGREEN modules have higher correlations and significance with the phenotype. MM.TAN was negatively correlated with AD, T3, and T4, and positively correlated with PRL, CA, Ca, and ZJ. The MM.YELLOWGREEN module adjusted DM, MD, and *p* upward, and it adjusted CP and Ca downward. The MM.DARKGREEN module was significantly positively correlated with T3, T4, and CP, and it significantly negatively correlated with DM (Figure 8C). Enrichment analysis was performed on genes from significantly correlated modules to identify the genes associated with skin development. The hub gene of the MM.DARKTURQUOISE module in WGCNA was *AMD5*, *Krt13*, and *TGM5* (Figure 8D).

#### 3.6.7. Quantitative Validation of Transcriptome Sequencing Results

In this study, we randomly selected six genes for qRT-PCR, thus verifying the accuracy of RNA-seq data, including *DSP* (Figure 9A), *CDH1* (Figure 9B), *PKP1* (Figure 9C), *Krt6a* (Figure 9D), *AOX1* (Figure 9E), and *FA2H* (Figure 9F) genes. The quantitative gene expression levels showed similar expression trends to RNA-seq data except for *AOX1* (Figure 9E), thus indicating that the accuracy of RNA-seq data was plausible.

## 4. Discussion

Adult chicken feathers undergo a cyclical development pattern, consisting of anagenesis, regression, and rest. Normal feather growth and regeneration play a crucial role in improving the welfare and economic value of poultry [18]. However, normal laying hens typically experience a loss of 10% to 15% of their feathers by 40 weeks of age [19]. The chickens involved in this experiment were 320 days old, and it was common for them to exhibit feather damage and bare skin before fasting. Subsequently, under the stimulation of fasting and light, the chickens underwent feather molting. The presence of complete feather coverage after feather molting is an indication of the experiment’s success in developing an effective model for inducing feather change. It also increases the value of cull hens. Nevertheless, characteristics pertaining to feather appearance, such as feather color, luster, and morphology, serve as significant reference indicators for determining the price of high-quality chickens [20]. The health and nutritional status of chicken feathers can be reflected in the physical properties of feathers and the feather’s chemical composition. The main component of feathers is the protein (89–97%), which contains a large amount of keratin. Keratin has been proven to be an effective adsorbent for specific harmful pollutants, owing to its large surface area and functional groups [21]. Therefore, the crude protein content of shed feathers is the highest [22]. In addition, feathers are expected to be valuable raw materials for the textile, plastic, cosmetic, pharmaceutical, biomedicine, bioenergy, and fertilizer industries. Their utilization in these industries has the potential to alleviate environmental pollution [23,24].

However, biosecurity issues such as avian influenza and novel coronavirus pneumonia have had a significant impact on the poultry industry’s development. As a result, certain regions have imposed restrictions on the sale of live chickens and instead encourage the trade of chilled chickens [25]. Previously, the quality of live chickens was assessed based on the condition of their feathers, whereas chilled chickens were evaluated by examining the appearance of the carcass [26]. This includes consumers evaluating carcass quality by observing factors such as skin thickness, pore density, and pore diameter [27].

According to a survey, chilled chicken with fine skin pores, a tender complexion, and good homogeneity was more likely to be popular among consumers [28]. The fine and dense pores after IM were expected to increase the economic value of marketed cull hens compared with before the trial.

Studies on HF have primarily focused on mammals. For instance, utilizing transcriptome and WGCNA in combination with differential gene screening, researchers were able to identify candidate genes that impact the fineness of the HF in velvet goats [29,30]. The HF trait in poultry has received less research attention. Ji et al. used microarray technology to screen candidate genes and molecular pathways associated with follicular traits in chicken skin feathers [31]. The differentiation of keratin-forming cells in avian scales was defined via transcriptomics [32]. By combining the phenotypes and existing research, in this experiment, we ultimately identified F0, F15, R5, and R32 as the time points for transcriptome sequencing.

Furthermore, the present study is a pioneering effort to comprehensively investigate the differential genes and pathways associated with skin follicle growth using three different methods. Among these genes, mucin 1 (*CDH1*), also referred to as epithelial cadherin (E-cadherin) [33], is known to promote cell adhesion and has been implicated in the prognostic assessment of oral squamous cell carcinoma using immunohistochemistry [34].

Mouse studies conducted in vitro and in vivo have highlighted the importance of CDH1 in inducing the dedifferentiation of keratin-forming cells and its contribution to skin regeneration triggered by mechanical stretch [35]. Moreover, the bridging granule protein (encoded by the *DSP* gene), which is highly abundant in myocardial tissue, regulates the transcription of adipose and fibrotic genes [35]. Recessive mutations in this protein disrupt the binding of bridging granules to epidermal keratin and muscle-specific intermediate filaments. As a result, individuals with these mutations may experience cardiomyopathy, skin fragility, and hair abnormalities [36,37].

We performed a correlation analysis between transcriptome data and phenotypic data and identified several modules that showed a strong correlation with T3 and T4 phenotypic traits. The hypothalamic–ovarian–gonadal axis plays a vital role in regulating hepatic lipid metabolism and maintaining glucose stability during periods of starvation, primarily through the influence of thyroid and sex hormones [38]. During IM, there is a significant decrease in plasma levels of FSH, LH, PRL, E2, and P4. However, levels of T3 and T4 significantly increase. These hormone levels gradually return to pre-replacement levels during the recovery period, playing a role in the regulation of feather replacement [11]. Thyroxine affects feather molting by enhancing the metabolic activity of feather-forming cells in a permissive rather than causative manner. However, the precise mechanism underlying this process has received limited research attention [39]. Further analysis of the modules that exhibited significant correlations with the phenotypic data revealed that the calcium signaling pathway may play a crucial role in fasting-induced feather replacement. Through pathway analysis, it was observed that starvation acts as a stimulus that triggers the pituitary gland to release TSH. Subsequently, substantial amounts of T3 and T4 were secreted by the thyroid gland in response to this hormone (Figure 10A). This, in turn, stimulated cellular calcium ion activity during the process (Figure 10B). The activation of the calcium signaling pathway leads to the activation of classical protein kinase Cα (PKC), which plays a role in the thyroid hormone signaling pathway. Additionally, PKC activation is involved in the activation of *Wnt*, *β-action*, *BMP*, and other pathways associated with skin follicle development (Figure 10C). The *PRKCB* gene, which encodes the PKC protein, has been identified as a differential gene in transcriptome data. In various biological processes, including human systemic lupus erythematosus, the regulation of autophagy, and gastric cancer, this gene plays a crucial role [40,41,42].

## 5. Conclusions

In this study, we observed and recorded feather and HF growth patterns during IM in more detail. In combination with transcriptome sequencing, three analyses were conducted to identify the key genes associated with the development of HF during IM. These genes include *DSP*, *CDH1*, *PKP1*, etc. Furthermore, we found that thyroid hormone may initiate feather Replacement by acting on the *PRCKB* gene in the calcium ion signaling pathway. Notably, fasting-induced feather change resulted in finer and denser pores. This development is anticipated to enhance the economic efficiency of the chilled chicken market.

## Figures and Tables

**Figure 1 biology-12-01345-f001:**
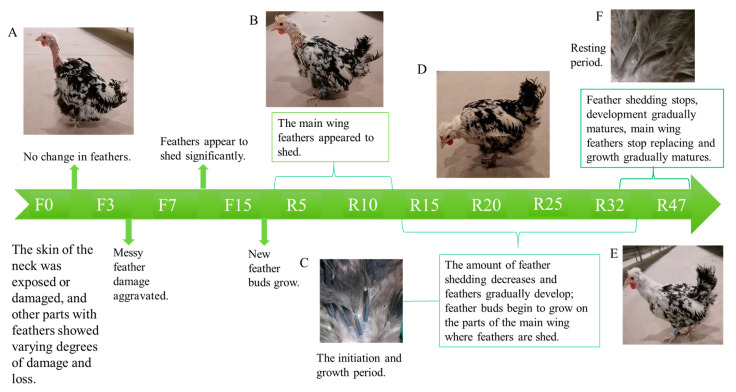
Feather changes during IM: (**A**) F0 period’s feather state; (**B**) R10 period’s feather state; (**C**) feathers in the initiation and growth phases; (**D**) R28 period’s feather state; (**E**) R47 period’s feather state; (**F**) resting feathers.

**Figure 2 biology-12-01345-f002:**
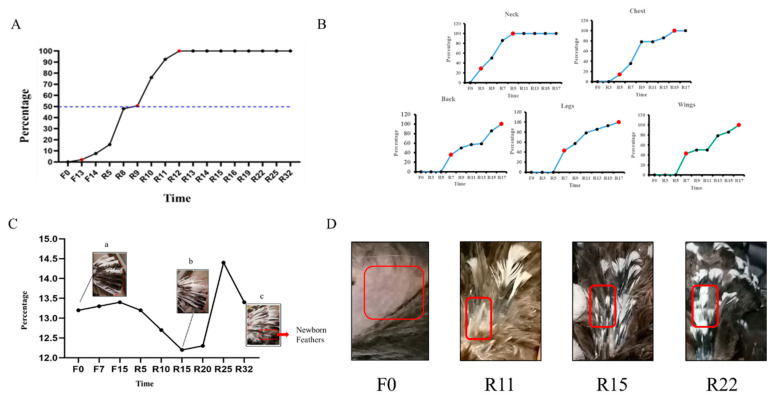
The statistics of feather shedding and regeneration: (**A**) shedding rate of late-laying hens at different periods; (**B**) regeneration of feathers in different parts of late-laying hens at different times; (**C**) variations in the length of the primary flight feathers at different periods in late-laying hens ((a,b,c) represent the changes in the main wing feathers during the F0, R15, and R32 periods, respectively); (**D**) induced changes in dorsal feathers during molting. In the red frame is an observation of the process of regenerating the feathers on the back.

**Figure 3 biology-12-01345-f003:**
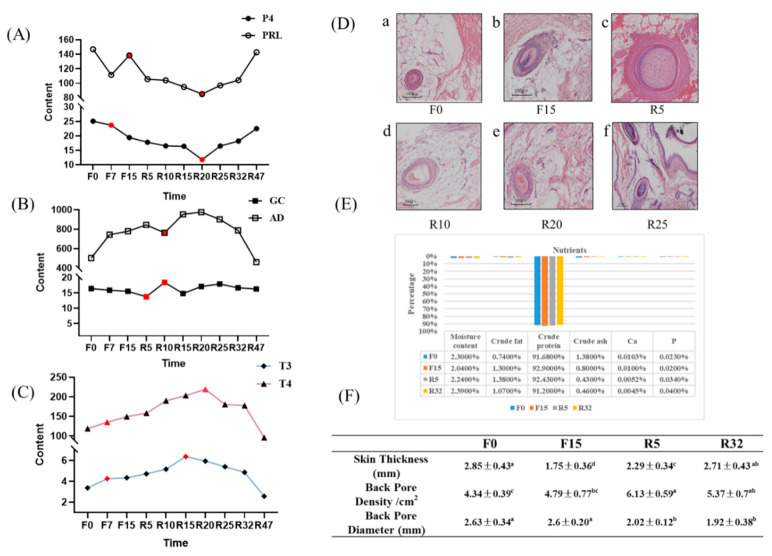
Hormone content changes, section observation, whole-body feather nutrient composition, and hair follicle traits: (**A**) changes in sex hormone content; (**B**) changes in hormone levels related to the adrenal axis; (**C**) changes in the levels of thyroid hormones; (The red dots represent periods when hormone levels vary significantly.); (**D**) the cross-sectional view of the dorsal skin HF at different periods; F0, F15, R5, R10, R20, and R25 are time points a–f, respectively. (**E**) results of the determination of each feather component in the periods of F0, F15, R5, and R32; (**F**) dorsal skin thickness, hair follicle density, and diameter at different periods of induced dorsal skin exchange. Different letters indicate significant differences (*p* < 0.05).

**Figure 4 biology-12-01345-f004:**
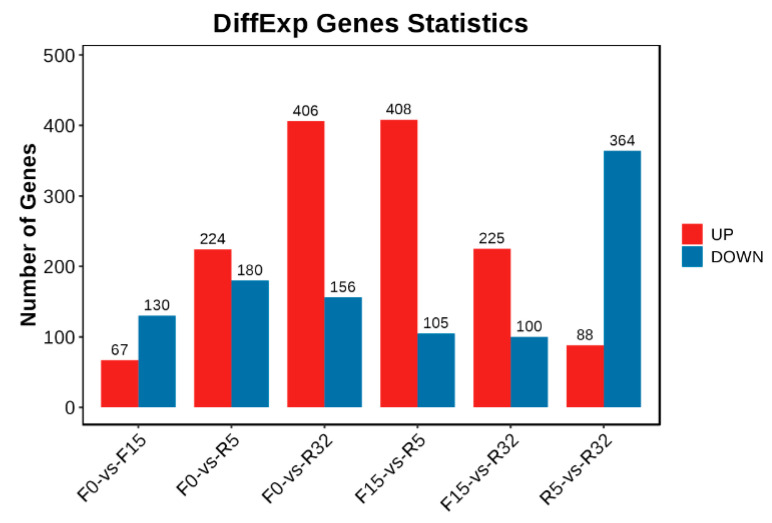
Histogram of the number of DEGs. Red represents an upward adjustment, and blue represents a downward adjustment.

**Figure 5 biology-12-01345-f005:**
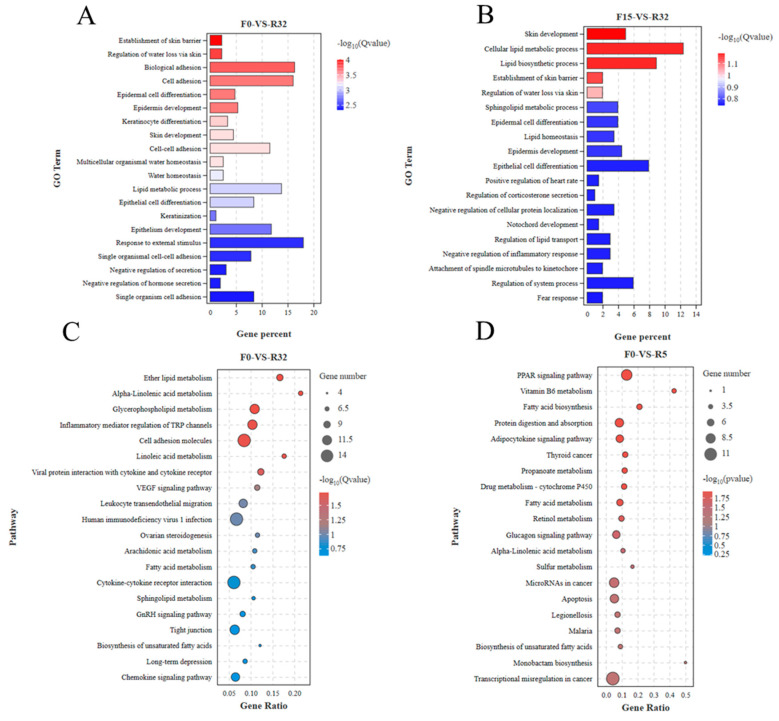
GO and KEGG enrichment analysis of differentially expressed genes: (**A**) GO of F0-VS-R32; (**B**) GO of F15-VS-R32; (**C**) KEGG of F0-VS-R32; (**D**) KEGG of F0-VS-R5.

**Figure 6 biology-12-01345-f006:**
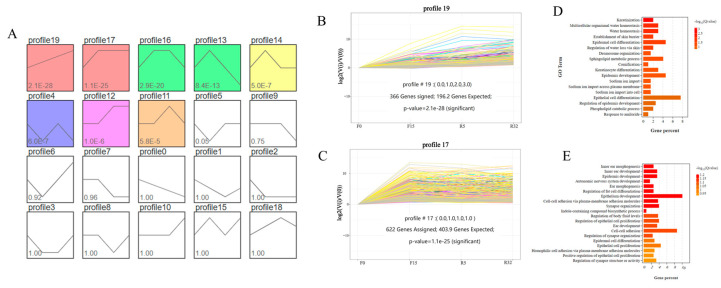
Trend analysis of differential genes: (**A**) 20 units based on *p* < 0.05; (**B**) unit 19 with 336 differential genes, *p*-value = 2.1 × 10^−28^; (**C**) unit 17 with 622 differential genes, *p*-value = 1.1 × 10^−25^; (**D**) GO analysis of differential genes in module 19; (**E**) GO analysis of differential genes in module 17.

**Figure 7 biology-12-01345-f007:**
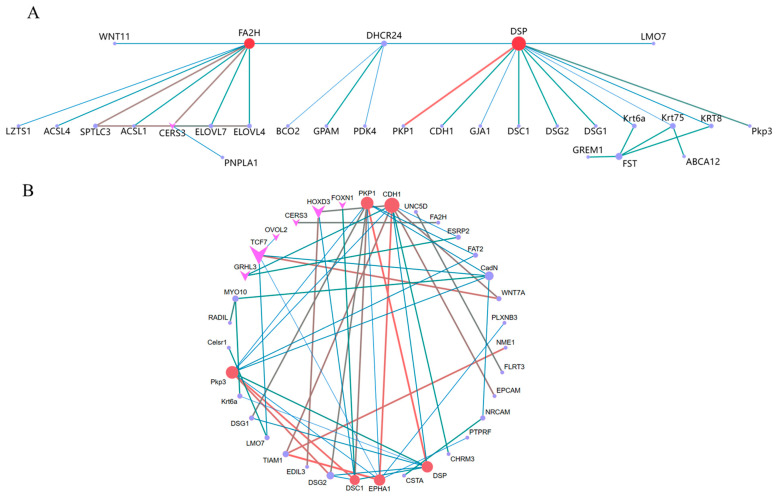
Differential analysis and series test of cluster analysis for gene PPI and gene regulatory network map of WGCNA: (**A**) analysis of variance; (**B**) series test of cluster. Arrows indicate transcription factors. The larger the circle and the darker the color, the stronger the connectivity.

**Figure 8 biology-12-01345-f008:**
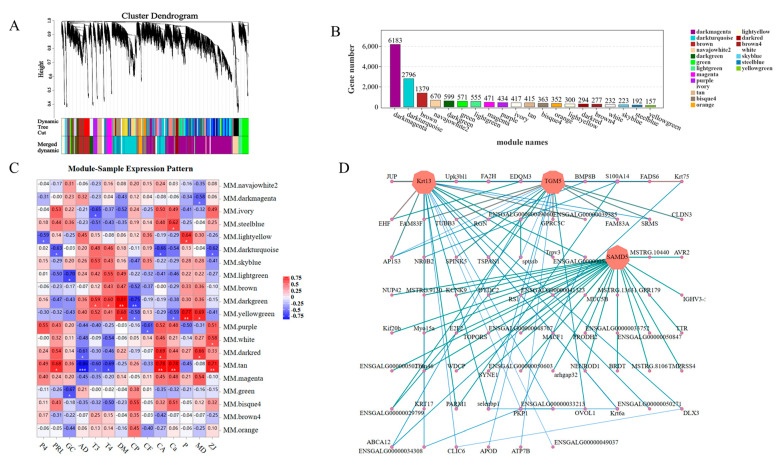
WGCNA analysis of skin follicle differential genes: (**A**) hierarchical clustering tree showing co-expression modules identified via WGCNA, with each leaf on the tree representing a gene and the main branches consisting of 19 modules marked with different colors; (**B**) the genes contained in each module; (**C**) module eigenvalues were correlated with different phenotypic data in correlation analysis (MD indicates skin density; ZJ indicates skin diameter) * *p* < 0.05, ** *p* < 0.01, *** *p* < 0.001; (**D**) the interactions of skin follicle-associated genes in a co-expression network in the selected MM.DARKTURQUOISE modules; the darker the color, the stronger the connectivity.

**Figure 9 biology-12-01345-f009:**
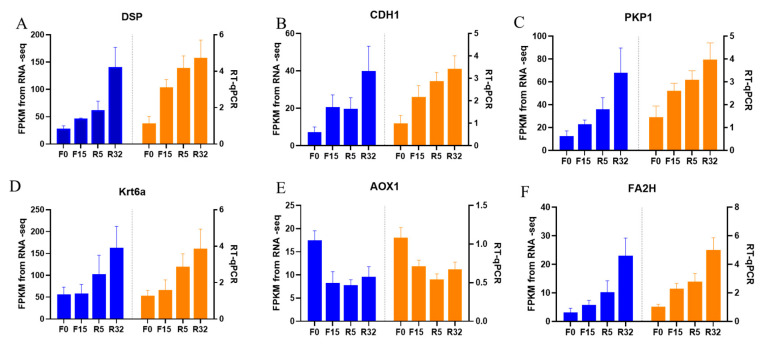
RT-qPCR validation of transcriptome data: (**A**) *DSP*; (**B**) *CDH1*; (**C**) *PKP1*; (**D**) *Krt6a*; (**E**) *AOX1*; (**F**) *FA2H*. Blue was the transcriptome FDKM value and orange was real-time fluorescence quantification.

**Figure 10 biology-12-01345-f010:**
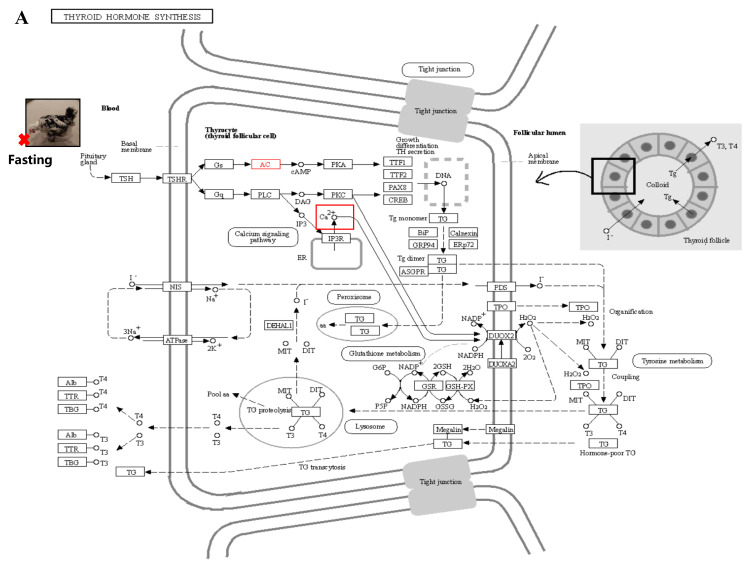
Signal path analysis: (**A**) thyroid hormone synthesis (Calcium ions within the cell are transported outside the cell via the transmembrane, as indicated by the red boxes.); (**B**) calcium signaling pathway; In the red box, PKC protein and calcium signal oh that pathway is involved in other pathways. (**C**) thyroid hormone signaling pathway. Red boxes represent pathways involved in hair follicle development.

**Table 1 biology-12-01345-t001:** Implementation plan of egg starvation method to IM.

Test Period	Processing Phase (Days)	Feed	Water	Light (h/d)
Fasting Period	1–3	No feed	×	8
	4–15	No feed	√	8
Recovery Period	16–30	Gradually resuming feeding	√	Increase 0.5 h per day until 16 h
Second egg-laying Period	30–49	Normal feed	√	16

Note: The symbol “×” indicates the absence of drinking water, while the symbol “√” represents normal drinking water.

**Table 2 biology-12-01345-t002:** Primer sequence data.

Target Gene	Primer Sequences (5’ to 3’)
*GAPDH*	F: GAACATCATCCCAGCGTCCA
	R:CGGCAGGTCAGGTCAACAAC
*DSP*	F: AAAGCAGGCTCTGGAGGCAT
	R: TTCCAGGCGTTGCTTCAAAC
*Krt6a*	F: ATGCAGACCCAGATCTCCGA
	R: GCAGCTCTTCGTACTTGGTT
*CDH1*	F: CTGTCTTCGTGCCCCCTATC
	R: CCCATGCGGTACGTGATCTT
*FA2H*	F: TCTTCCACATGAAGCCACCC
	R: TGTCAAAGGGGGACTTGTGG
*AOX1*	F:CGTGAATGGGAGAAAGGTGGT
	R: ACCTCCTCCACAGCCATACT
*PKP1*	F: TCATGTCCAACCCACACCTG
	R: TGCTTTTGGACAGGAGCCAT

## Data Availability

The datasets from the current study are available from the corresponding author upon reasonable request. Transcriptomic data of the dorsal skin of 12 guinea hens are available from NCBI (https://dataview.ncbi.nlm.nih.gov/object/PRJNA988236?reviewer=rg5cpo55kctjhclc7ih3uk80pa) (accessed on 12 August 2023).

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
