# Peer review of "Multi-Stage Transcriptome Analysis Revealed the Growth Mechanism of Feathers and Hair Follicles during Induction Molting by Fasting in the Late Stage of Egg Laying"

_biology, 2023, doi:10.3390/biology12101345_

Round 1

Reviewer 1 Report

The writing and language usage is moderate overall. While the paper communicates the key ideas, there are a number of grammatical, spelling, punctuation, and word choice issues throughout that need to be addressed.

Reviewer 2 Report

Dear researchers:

I consider that the work carried out was very complete and with many complications. I consider it very complex and even at one point in the paper one loses between the initial objective and the results, I feel that there is a lack of writing in that part. It is not clear to me the fact of defining the DSP, CDH1, PKP1 genes in relation to the development of heart failure during MI, I think there is also a lack of wording. Regarding the information provided and the figures and charts posted here, I consider that they are sufficient to support the development of the experiments.

My additional and specific comments such as:

1. What is the specific mechanism for feather replacement and the most typical epigenetic signs of physiological remodeling induced by fasting?

2. It provides potential tools for feather molting through thyroid hormone synthesis, calcium signaling, and thyroid hormone signaling pathways. The above could improve the economic efficiency of the chilled chicken market

3. Analyzes to identify key genes associated with the development of heart failure during MI. These genes include DSP, CDH1, PKP1, etc. Additionally, the study found that thyroid hormone can initiate feather replacement by acting on the PRCKB gene in the calcium ion signaling pathway.

4. Comparison with different types of shedding. Add groups to determine egg characteristics (egg weight, shell thickness and numbers of eggs per cycle.

5. Conclusions are consistent with the evidence and arguments presented and they address the main question posed

6. references are appropriate

7. I consider that tables and figures are adequate to emphasize the results of the experiment, although in some cases they are too schematic. I recommended that this kind of question will be included in the initial and general format.

Round 2

Reviewer 1 Report

I am happy with the revised manuscript; however, the authors provided information (log2fc, pvalue, and FDR) for only the top 10 genes, and none of the genes such as DSP, CDH1, and PKP1 are in any of the groups. The authors should update Table S2 with all DEGs for all groups in their proofreading version.

In my opinion, the current manuscript is acceptable for the biology journal.